# Tailoring sodium intercalation in graphite for high energy and power sodium ion batteries

Zheng-Long Xu[1], Gabin Yoon[1], Kyu-Young Park[1], Hyeokjun Park[1], Orapa Tamwattana[1], Sung Joo Kim[1], Won Mo Seong[1] & Kisuk Kang[1]

Co-intercalation reactions make graphite as promising anodes for sodium ion batteries, however, the high redox potentials significantly lower the energy density. Herein, we investigate the factors that influence the co-intercalation potential of graphite and find that the tuning of the voltage as large as 0.38 V is achievable by adjusting the relative stability of ternary graphite intercalation compounds and the solvent activity in electrolytes. The feasibility of graphite anode in sodium ion batteries is confirmed in conjunction with $Na_{1.5}V$-$PO_{4.8}F_{0.7}$ cathodes by using the optimal electrolyte. The sodium ion battery delivers an improved voltage of 3.1 V, a high power density of 3863 W kg$^{-1}$ $_{both\ electrodes}$, negligible temperature dependency of energy/power densities and an extremely low capacity fading rate of 0.007% per cycle over 1000 cycles, which are among the best thus far reported for sodium ion full cells, making it a competitive choice in large-scale energy storage systems.

[1] Department of Materials Science and Engineering, Research Institute of Advanced Materials (RIAM), Seoul National University, 1 Gwanak-ro, Gwanak-gu, Seoul 151-742, Republic of Korea. Correspondence and requests for materials should be addressed to K.K. (email: matlgen1@snu.ac.kr)

Driven by the concerns of limited and unevenly distributed lithium resources, sodium ion battery (SIB) technologies have taken the privilege over lithium ion batteries (LIBs) on meeting the emerging and demanding applications of large-scale energy storage systems, because of the abundant sodium resources in the earth crust and sea water[1–3]. During the past few years, great efforts have been devoted to exploring suitable electrode materials to take advantage of the cost-effective sodium-rechargeable battery chemistry[4,5]. In the endeavor, metal alloys/oxides/chalcogenides, phosphorus, and carbonaceous materials have been extensively studied as potential anodes for SIBs, showing great promises[6–11]. In particular, materials undergoing alloying and/or conversion reactions could present high sodium storage capacities and low operation potential, which could benefit the energy density of SIBs. Moreover, traditional carbonaceous materials such as hard carbon were capable of delivering a respectable cyclic stability with high capacities of ~300 mAh g$^{-1}$ [12,13]. Nevertheless, the drastic volume changes that are typically accompanied in the alloying or conversion reactions cause the structural degradation of electrode materials, leading to a relatively fast capacity fading[9,10]. Regarding the use of the hard carbon anode, the low redox voltage close to that of the sodium metal plating[12] would induce serious concerns on the safety issues.

Graphite has been recently revisited as a potential anode material for SIBs. At the early stage of the research, it was often demonstrated that only a negligible amount of sodium ions can be intercalated into the graphite structure (i.e., 30 mAh g$^{-1}$), which was attributed to the thermodynamic instability of the binary Na-intercalated graphite compound (b-GIC) based on computational studies[14,15]. Meanwhile, it was revealed by Jache et al.[16] and our group[17] that sodium can be reversibly stored in graphite through co-intercalation reactions, where solvated sodium ions are intercalated into the galleries of graphite, forming a ternary graphite intercalation compound (t-GIC). Co-intercalation reactions endowed graphite electrodes with a remarkable reversibility and fast sodium intercalation kinetics, opening the new avenue toward exploiting graphite as a promising anode for advanced SIBs.

Motivated by these findings, the electrochemical performance of the graphite anode and its intrinsic reaction mechanisms have been extensively investigated in recent years. Yoon et al.[18] and Jung et al.[19] described the specific conditions that trigger the co-intercalation reactions in the electrochemical cells based on density functional theory (DFT) calculations. Goktas et al.[20] found that several solvents like crown ether that were unsuitable for the co-intercalation reaction at room temperature could be activated at elevated temperatures. It was also unveiled that the co-intercalation in graphite occurs via a fast staging process[21,22], with the growth of a very thin solid electrolyte interphase (SEI) layer. The structural stability and the fast kinetics led to the experimental realizations that the graphite anodes consistently display the cyclic stability up to thousands of cycles and high rate capabilities[23,24]. When assembled in the sodium ion full cell using Na$_{0.7}$CoO$_2$ as a cathode, it could still retain a high-power capability of 300 W kg$^{-1}$ and long cycle life of 1000 cycles[25]. Nonetheless, one of the grand challenges in exploiting the graphite anode till now has been its relatively high co-intercalation redox potential. According to the sodium ion full cell demonstrated by Hasa et al.[25], the output voltage was only 2.2 V due to the high potential exhibited by graphite anode (i.e., 0.77 V), which led to an unsatisfactory energy density of 60 Wh kg$^{-1}$, far from sufficient to meet the requirement for practical applications. In other series of reports, the sodium co-intercalation voltages were also identified in the range of 0.6−0.8 V vs. Na$^+$/Na using various ether-based electrolytes[21,22,26]. These values are far higher than the ~0.15 V vs. Li$^+$/Li for the conventional intercalation reactions of lithium to the graphite anode in LIBs. How to tailor the co-intercalation potential for SIBs with high output voltage that can lead to high energy and power densities still remains an open question.

In this work, we aim to tune the sodium intercalation potential in graphite by systematically investigating the correlations among the co-intercalation voltage, the solvent species, the concentrations of sodium ions in the electrolytes, and the temperatures. It is found that the relative stability of the ternary graphite intercalation compounds and the free solvent activity are one of the most dominant factors determining the sodium storage voltage. Following these insights, a large tuning range of 380 mV is obtained with the lowest reaction voltage of 0.43 V for graphite-based half-cells. When coupling the graphite anodes with Na$_{1.5}$VPO$_{4.8}$F$_{0.7}$ cathodes and an optimal electrolyte, we attain Na-ion full cells presenting one of the highest power densities with the output voltages of 3.1 V and impressive capacity retention at a wide range of temperature (from 0 to 60 °C), which are among the best thus far reported for Na-ion full batteries, promoting the practical application of secondary Na-ion batteries with feasible graphite anodes.

## Results

**Co-intercalation voltage of graphite anode in SIB.** In tailoring the co-intercalation reaction potential, it is necessary to recall the relevant reaction processes involving sodium, solvent, and graphite. The electrode reactions for a half-cell of graphite|| 1 M NaPF$_6$ diethylene glycol dimethyl ether (G$_2$)|| sodium can be described as follows:

$$\text{Anode}: \text{Na} + (\text{G}_2) \leftrightarrow [\text{Na} - \text{G}_2]^+ + e^- \quad (1)$$

$$\text{Cathode}: x\,\text{C} + [\text{Na} - \text{G}_2]^+ + e^- \leftrightarrow [\text{Na} - \text{G}_2]\text{C}_x \quad (2)$$

where $[\text{Na} - \text{G}_2]^+$ refers to the solvated Na ions and $[\text{Na} - \text{G}_2]\text{C}_x$ is the t-GIC (Fig. 1a). Based on Eqs. (1) and (2), the co-intercalation reaction voltage (V) can be expressed as

$$V = (E^0_{t-\text{GIC}} - E^0_{\text{Na}}) + \frac{2.303RT}{nF}\log a_{\text{G2}} \quad (3)$$

where $E^0_{t-\text{GIC}}$, $E^0_{\text{Na}}$, $R$, $T$, $F$, and $a_{\text{G2}}$ denote the standard electrode potentials of t-GIC and Na metal, gas constant, absolute temperature, Faraday constant, and the solvent activity, respectively. Accordingly, the co-intercalation voltage depends on the standard formation potential of t-GIC $\left(E^0_{t-\text{GIC}}\right)$ and the activity of free solvent molecules ($a_{\text{G2}}$).

**Relative stability of ternary GIC.** In order to understand the effect of these energetics, we conducted a series of co-intercalation experiments using six different linear solvent species with increasing average chain lengths (i.e., dimethyl ether (G$_1$), G$_2$, triethylene glycol dimethyl ether (G$_3$), tetraethylene glycol dimethyl ether (G$_4$), and poly (ethylene glycol) dimethyl ether (G$_{n-250}$ with $M_w \approx 250$, and G$_{n-500}$ with $M_w \approx 500$)). Supplementary Fig. 1 presents the charge/discharge profiles of graphite cycled in 1 M NaPF$_6$ electrolytes with the respective solvents at 50 mA g$^{-1}$ in the voltage range of 0.1–2.0 V. The shape of the charge/discharge profiles and the Na storage capacities in G$_1$, G$_2$, G$_4$, and G$_n$ systems are similar, where the specific capacities are in the range of 105–120 mAh g$^{-1}$. It is noted that the co-intercalation of graphite in G$_3$ system is only partly reversible; thus, we will exclude the G$_3$ system in the following discussion on electrochemical performance[22]. Figure 1b illustrates the differential capacity versus voltage (dQ/dV) curves derived from the charge/discharge profiles. It clearly presents that the average Na

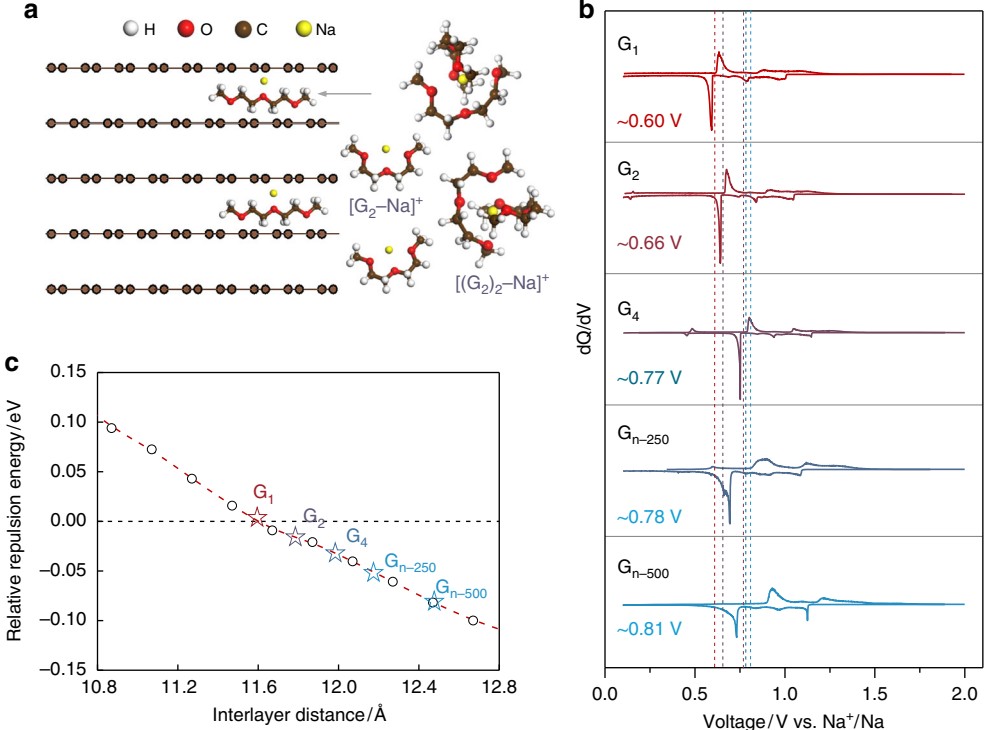

**Fig. 1** Solvent dependency of co-intercalation voltage. **a** Schematic illustration of solvated Na ion co-intercalating into graphite galleries. **b** dQ/dV plots of graphite electrodes cycled in 1 M NaPF$_6$ G$_1$, G$_2$, G$_4$, G$_{n-250}$, and G$_{n-500}$ electrolytes, respectively. **c** Plot of normalized repulsion energy as a function of interlayer distance to elucidate the correlation between Na storage potential in graphite and ethers used in electrolytes

co-intercalation voltage alters systematically from 0.60, 0.66, 0.77, and 0.78 to 0.81 V for G$_1$, G$_2$, G$_4$, G$_{n-250}$, and G$_{n-500}$-based electrolytes, respectively. It suggests an increase of the average co-intercalation voltage, as the chain length of the solvent increases. Note that the average co-intercalation voltage is determined from dQ/dV curves by the dashed lines located between discharge and charge plateau potentials (Fig. 1b), similar with previous studies[16,17,20]. Also note that the average co-intercalation voltages are consistent with the values calculated by the method of dividing the energy density into specific capacity.

The $E_{t-GIC}^0$ depends on the thermodynamic stability of t-GICs, which can be considered with respect to the following interactions, namely, (i) the interaction between the intercalants and graphene layers, (ii) the repulsion among positively charged intercalants, and (iii) the effective repulsion among negatively charged graphene layers[27]. As the chain length of the ether solvent increases, it is expected that the Na ions are more efficiently shielded in the solvated structure; therefore, the unfavorable interaction between the Na intercalant and the graphite, which has prevented the intercalation of the bare Na ion in the graphite structure[18,27], would be significantly weakened. It would lead to the stabilization of the t-GIC structure, and accordingly, the average co-intercalation voltage would increase with longer ether solvent molecules. As the Na ions are better shielded in the solvated structure, the repulsion among positively charged intercalants is also expected to decrease as well, which aids in the further stabilization of the t-GICs[21]. Finally, the repulsive interaction between the negatively charged graphene layers would be mitigated by the larger interlayer distances offered by the co-intercalation. The interlayer distances calculated from XRD patterns of the fully sodiated graphite (Supplementary Fig. 2) present the values of about 11.6, 11.8, 12.01 12.2, and 12.5 Å for G$_1$, G$_2$, G$_4$, G$_{n-250}$, and G$_{n-500}$ systems, respectively[22]. When the relative stabilities of the graphite host are estimated as a

function of interlayer distance from the DFT calculations as depicted in Fig. 1c and Supplementary Fig. 2, it is observed that the repulsion energy decreases with the distance between graphene layers. We would like to note that the amount of charge transferred from the intercalants to graphene layers is almost identical among co-intercalated t-GICs, because of their similar reversible capacities (Supplementary Fig. 1) and Bader charges of intercalants (Supplementary Table 1). This allows us to compare the repulsive interactions of all compounds in a single curve in Fig. 1c. As indicated with stars in the dashed line, the weaker repulsion between graphene layers stabilizes the t-GIC structure, resulting in higher intercalation potentials, with the longer chain of the ether solvents. It illustrates the efficacy of regulating the co-intercalation potential by a proper selection of solvents.

**Solvent activity dependency of co-intercalation voltage.** To further investigate the effect of solvent, we systematically altered the solvent activity ($a_{G2}$ of a specific solvent (G$_2$-based electrolyte)) by preparing a series of electrolytes with different salt concentrations (NaPF$_6$ from 0.05 to 3 M). The charge/discharge profiles of graphite electrodes cycled in these electrolytes at 50 mA g$^{-1}$ are presented in Supplementary Fig. 3. Figure 2a depicts that the Na storage voltages are apparently shifted to lower values, as the concentration of electrolytes increases. The correlation between the concentration and the Na half-cell voltage is more clearly displayed in Fig. 2b. In the relatively diluted region from 0.05 to 1.0 M, only a slight shift of the co-intercalation voltage is observed from 0.630 to 0.622 V. On the other hand, a more abrupt change in the voltage is accompanied with the concentration increase from 1.0 to 3.0 M, suggesting the efficacy of highly concentrated electrolytes to tune the voltage.

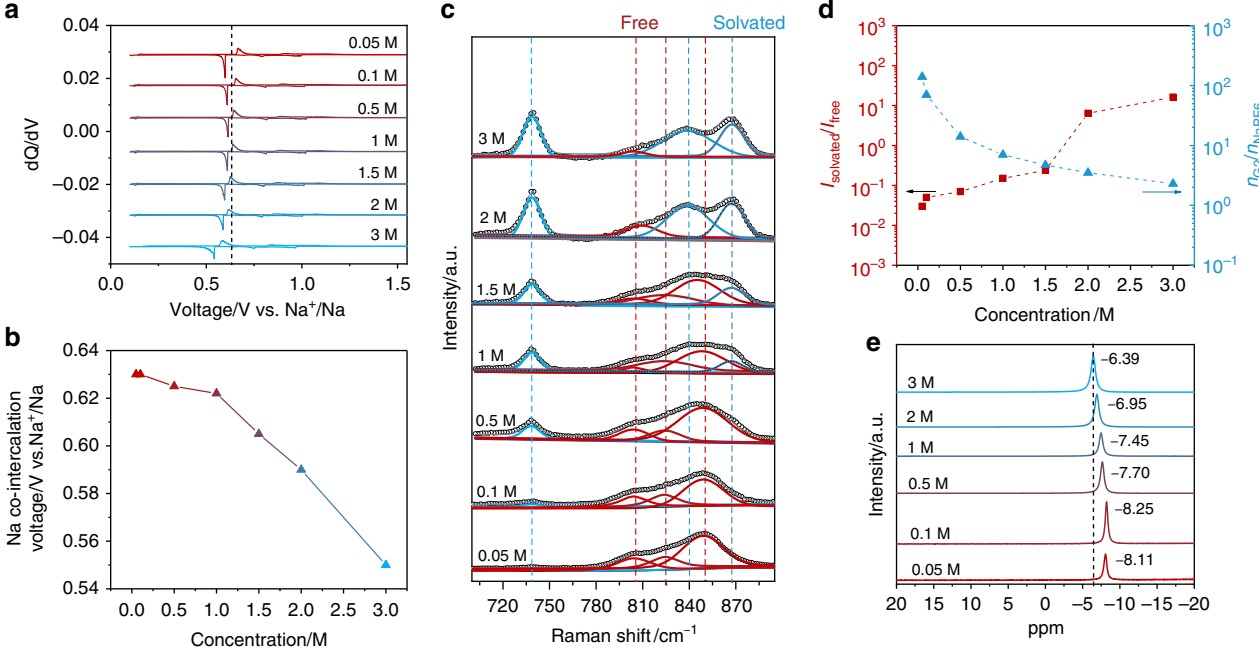

**Fig. 2** Co-intercalation voltage change with electrolyte concentrations. **a** $dQ/dV$ curves of graphite anodes cycled in $NaPF_6$ $G_2$ electrolytes with concentrations from 0.05 to 3 M at 35 °C. **b** Average co-intercalation voltages in Na half-cells derived from (**a**) plotted as a function of electrolyte concentration. **c** Deconvoluted Raman spectra of electrolytes for different concentrations. The peak at ~738 cm$^{-1}$ that corresponds to the P-F bond from $PF_6^-$ anion increases in the intensity monotonously with the salt concentration, implying that the electrolytes were well prepared for targeted compositions. **d** Intensity ratio of Raman peak between solvated solvent to free solvent and the molar ratio of $G_2$ solvent to $NaPF_6$ salt. **e** $^{23}Na$ NMR spectra of electrolytes as a function of concentration

In order to elucidate the negative shift and the better efficacy in higher concentrations, the structure of electrolytes was first examined by Raman. Figure 2c shows Raman spectra of the electrolytes with concentrations from 0.05 to 3 M (the full spectra are presented in Supplementary Fig. 4). By dissolving $NaPF_6$ salt in $G_2$ solvent, two new peaks at 839 and 867 cm$^{-1}$ referring to solvation of $Na^+$ by $G_2$ gradually appear, while the intensities of the free $G_2$ solvent peaks at 805, 825, and 850 cm$^{-1}$ decrease, implying the increased ratio of coordinated solvents[28]. The intensity ratios of coordinated solvent to free solvent molecules ($I_{Solvated}/I_{Free}$) are plotted as a function of the concentration in Fig. 2d. It illustrates that the ratio increases from 0.04 (at 0.05 M) to 0.23 (at 1.5 M) and 15.7 (at 3 M), while molar ratios of $G_2$ solvent to $NaPF_6$ salt drastically decrease from 142 (at 0.05 M) to 4.7 (at 1.5 M) and 2.3 (at 3 M). Both results suggest that a significant reduction of free solvent molecules takes place, particularly at a highly concentrated electrolyte above 1.5 M[29]. Fourier-transform infrared (FTIR) spectra in Supplementary Fig. 4 also support the similar trend of the decrease of the free solvent, consistent with the Raman analyses. In addition, the local environment of $Na^+$ in $G_2$ with different concentrations probed by $^{23}Na$ nuclear magnetic resonance (NMR) in Fig. 2e indicates the $^{23}Na$ chemical shift from $-8.11$ to $-6.39$ ppm when the concentration increased from 0.05 to 3 M. It is attributable to the increased intensity of contact ion pairs between the negative $PF_6^-$ anion and the $Na^+$ cation in the concentrated electrolyte[30], leading to a lower electron density of the $Na^+$ nucleus. These results consistently suggest the substantial reduction in the amount of free solvent molecules for concentrated $NaPF_6$-$G_2$ electrolytes.

Inspired by this, we estimated the activities (or activity co-efficiencies) of $G_2$ solvent in the concentrated electrolytes. The electromotive force (EMF) was measured by recording the potential difference between two Na metal specimens in the sample

electrolytes and the reference electrolyte (1 M $NaPF_6$ $G_2$), which were connected by a vycor glass[30]. The EMF increased linearly with logarithmic concentration ($\log c_{Na^+}$) below 1 M (Supplementary Fig. 5), as predicted by the Nernst equation. However, in the high-concentration region from 1 to 3 M, we found that the EMF increased more rapidly and was about 150 mV higher in the 3 M electrolyte than that for the reference. To interpret this notably high potential difference, we turned to the electrode potential ($E_{Na}$) of Na metal with solvation/desolvation reactions (see Eq. (1)):

$$E_{Na} = E_{Na}^0 + \frac{2.303RT}{F} \log \frac{a_{[Na-G2]^+}}{a_{G2}} \qquad (4)$$

where $E_{Na}^0$ is the standard electrode potential, $a_{[Na-G2]^+}$ and $a_{G2}$ refer to the activities of $[Na-G_2]^+$ and free $G_2$, respectively. Based on the Raman and FTIR analyses, the amount of free $G_2$ is extremely low in high-concentration electrolytes, yielding the $a_{G2}$ effectively near to zero. In Eq. (4), the logarithm diverges to an infinite value if $a_{G2} = 0$, thus qualitatively explaining the sharp increase of the electrode potential in the concentrated regime[30–32]. By combining these results, we can postulate that the free solvent activity in the electrolyte can be significantly reduced by preparing highly concentrated electrolytes, which in turn would decrease the average voltage for the co-intercalation reactions (see Eq. (3)). In order to support this, we additionally carried out similar sets of experiments using other electrolyte systems, i.e., $NaPF_6$ dissolved in $G_1$ from 0.1 to 2 M, $NaClO_4$ dissolved in $G_2$ from 0.5 to 3 M, and $NaCF_3SO_3$ dissolved in $G_2$ from 0.1 to 2 M (Supplementary Fig. 6). All exhibited a consistent trend of the negative shift of the co-intercalation voltage, with the increase of the salt concentration in electrolytes, further verifying the effectiveness of tailoring the co-intercalation voltage by regulating the solvent activity, particularly at high concentrations. Note that the co-intercalation voltage change could be related to the potential changes of both graphite

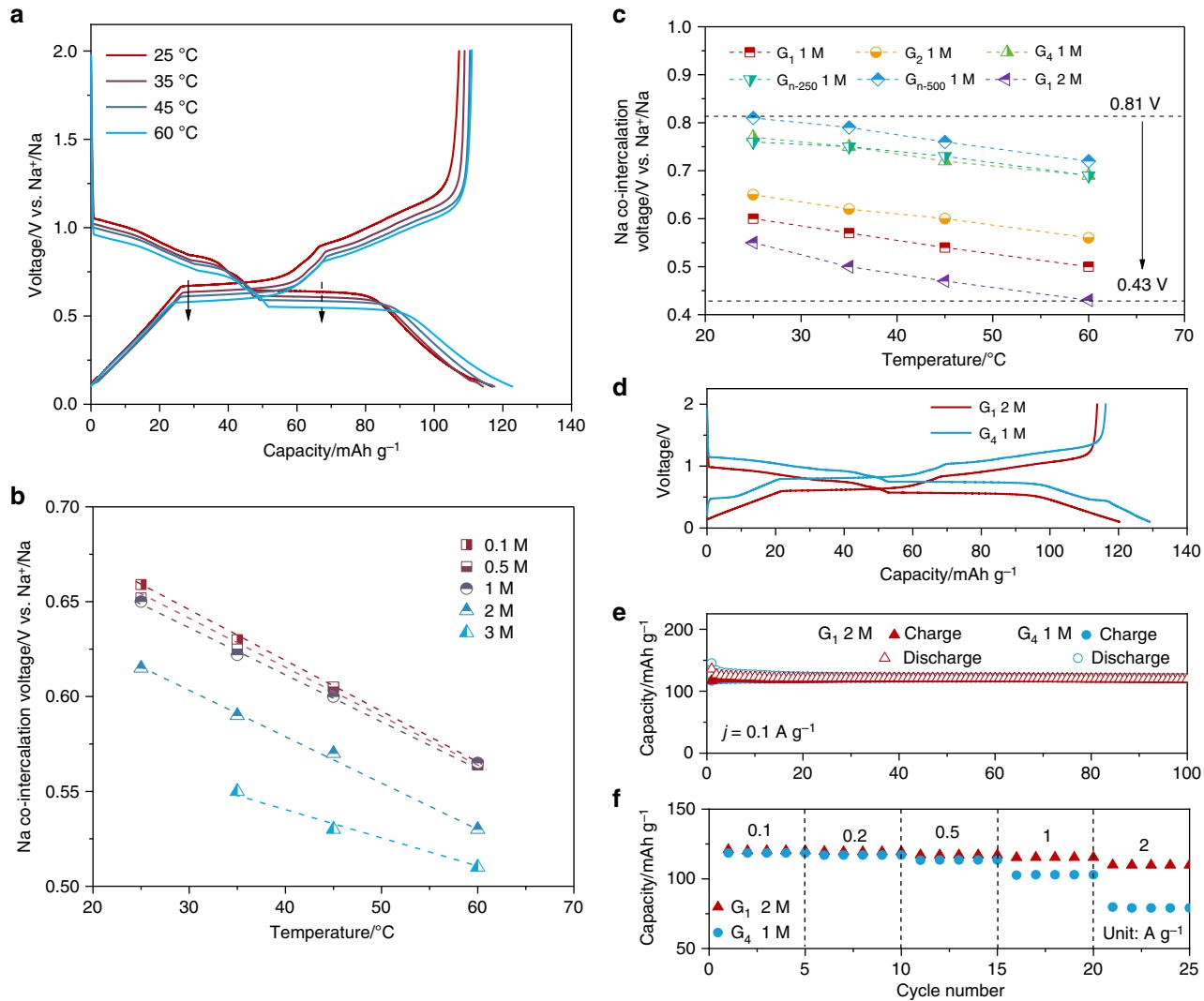

**Fig. 3** Temperature dependency of co-intercalation voltage and optimal conditions for low co-intercalation voltage. **a** Galvanostatic discharge/charge profiles of graphite anodes in 1 M NaPF$_6$ G$_2$ electrolyte at different temperatures. **b** Na co-intercalation voltage in graphite anodes at different concentrations of G$_2$-based electrolytes plotted as a function of operating temperature. **c** Summary of Na co-intercalation voltage at different temperatures in different electrolytes. **d** Discharge/charge voltage profiles. **e** Cyclic capacities and **f** rate capabilities of graphite anodes cycled in 1 M G$_4$ and 2 M G$_1$-based electrolytes

electrode and Na metal electrode (see the detailed discussion in Supplementary Note 1).

**Temperature dependency of co-intercalation voltage**. We observed an apparent negative shift of the electrode potentials for co-intercalation, when the graphite-based half-cells were cycled at elevated temperatures from 25, 35, and 45 °C to 60 °C (marked with black arrows in Fig. 3a). Figure 3b more clearly displays a temperature dependence of the co-intercalation voltage, which appears to be a linear relationship with negative slopes. The temperature coefficients ($\Delta E/\Delta T$) were calculated to be about $-2.85$ mV K$^{-1}$ for dilute electrolytes (0.1–1 M) and ` $-1.6$ mV K$^{-1}$ for high-concentration electrolytes (2 M and 3 M), respectively. The difference in the temperature coefficient for each electrolyte system is attributed to the low activity of free G$_2$ solvent, according to the following equation derived from Eq. (3):

$$\frac{\partial E}{\partial T} = \frac{2.303R}{nF}\log a_{G_2} \qquad (5)$$

In a viewpoint of classical thermodynamics, the temperature coefficient can be also expressed with entropy changes ($\Delta S$) of the co-intercalation reactions as follows:

$$\Delta S = nF\frac{\partial E}{\partial T} \qquad (6)$$

Compared with the temperature coefficient for lithium ion intercalation reaction in graphite ($-0.0512$ mV K$^{-1}$)[33,34], the $\Delta E/\Delta T$ values in this work are orders of magnitudes greater, suggesting the higher-entropy changes of sodium co-intercalation reaction over the conventional ion intercalation reactions. It is attributable to the generally greater difference in the entropy when involving liquid or gas-phase reactant in the overall reaction, such as solvent molecules in this case. Nevertheless, further studies are needed to elucidate the origin of the distinct $\Delta S$ behavior, depending on the intercalation mechanisms.

In Fig. 3c, we summarized the variations of the co-intercalation potentials for graphite electrode, considering solvent species, salt concentrations, and temperatures. It indicates that a substantial

co-intercalation voltage tuning as much as 380 mV is possible by properly adjusting the components of electrolytes and the operation temperatures. It is worth noting that the shift of co-intercalation potential is present by the plateau voltage change in half-cells. It is also worthwhile to note that the discernible shift of the redox potential is not generally achievable for conventional intercalation reactions. When the similar changes in the conditions are applied for the graphite electrode in the conventional lithium intercalation, no noticeable change in the intercalation potential was observed as demonstrated in Supplementary Fig. 7, suggesting that it is a unique property for the co-intercalation reactions. In Fig. 3d, as one of the examples of the combined effect of solvent species and the concentration, we comparatively display the electrochemical profiles of the graphite electrodes, using 2 M $NaPF_6$ $G_1$- and 1 M $NaPF_6$ $G_4$-based electrolytes at room temperature. It reveals that the average co-intercalation voltage is about 0.56 V in 2 M $G_1$-based electrolyte, which is notably lower than 0.78 V in 1 M $G_4$-based electrolyte in the same operating conditions. Despite the change in the redox potential, there was no noticeable difference either in sodium storage capacity or cycle stability for these two electrolyte systems as shown in Fig. 3e. However, it was found that the cell in 2 M $G_1$-based electrolyte exhibited a better power capability than that in 1 M $G_4$-based electrolyte (Fig. 3f). It is probably due to the lower viscosity (3.1 cp vs. 7.5 cp for $G_4$-based electrolyte) and higher ionic conductivity (16.6 mS vs. 2.3 mS for $G_4$-based electrolyte) of the 2 M $G_1$-based electrolyte. It is worth noting that we also prepared $G_1$-based electrolyte with higher concentrations (i.e., 2.5 and 3 M) to explore the limit of using a highly concentrated electrolyte to further decrease co-intercalation voltage. However, electrodes cycled in 2.5 and 3 M $G_1$-based electrolytes presented inferior rate capabilities (Supplementary Fig. 8), due to the high viscosity of the electrolytes. In total, 2 M $G_1$-based electrolyte was thus identified as close to the sweet spot for further electrochemical evolutions. We additionally conducted analogous experiments at 60 °C as presented in Supplementary Fig. 9, which showed similar results that the combined effects of the solvent species, concentrations, and temperature can cumulatively alter the redox potential, without the significant alternation in other electrochemical properties.

**Electrochemical performance of Na-ion full batteries**. To estimate the electrochemical performance of graphite in Na-ion full cells, $Na_{1.5}VPO_{4.8}F_{0.7}$ was prepared and used as a cathode (the details on the sample characterizations can be found in Supplementary Fig. 10)[35,36]. In a selected 2 M $G_1$-based electrolyte, the $Na_{1.5}VPO_{4.8}F_{0.7}$ cathode presented a reversible capacity of ~120 mAh g$^{-1}$, an average Na storage potential of 3.8 V (vs. Na$^+$/Na), and a respectable cyclic reversibility for 150 cycles at 50 mA g$^{-1}$ (Fig. 4a). The $Na_{1.5}VPO_{4.8}F_{0.7}$ cathodes were also tested in 1 M $G_1$, $G_2$, and $G_4$-based electrolytes at room temperature (25 °C) and 2 M $G_1$-based electrolyte at 60 °C, respectively (Supplementary Fig. 11). No apparent differences in charge/discharge capacities or average redox potentials were observed, indicating the independence of sodium storage behaviors for $Na_{1.5}VPO_{4.8}F_{0.7}$ cathodes on the electrolyte systems. In addition, the electrochemical stability of electrolytes used in this work was examined in Supplementary Fig. 12, which indicated that no significant oxidative reaction was observed up to 4.5 V[37,38], ensuring their suitable application in Na-ion full batteries.

The Na-ion full cells were assembled by using graphite anodes and $Na_{1.5}VPO_{4.8}F_{0.7}$ cathodes with optimal mass ratios. Precycling of the electrodes was conducted before testing[39]. The full cells cycled in 2 M $G_1$-based electrolyte delivered a capacity of

~120 mAg h$^{-1}$ based on the mass of an anode, with an average voltage of ~3.1 V at 100 mA g$^{-1}$ (Fig. 4b), leading to a high-energy density of 149 Wh kg$^{-1}$ based on the total mass of anode and cathode active materials. The well-defined high-voltage plateaus could be stably maintained over cycles with a respectable cyclic retention performance. On the other hand, graphite||$Na_{1.5}VPO_{4.8}F_{0.7}$ full cells cycled in 1 M $G_4$-based electrolyte presented a much lower average output voltage of 2.78 V (Fig. 4c), due to the relatively higher co-intercalation potential of graphite in the half-cell. The full cells were further cycled in other electrolyte systems or operating conditions such as 1 M $G_2$-based electrolytes at room temperature and 2 M $G_1$-based electrolyte at 60 °C (Supplementary Fig. 13). They presented average output voltages of 2.9 and 3.2 V, respectively, further evidencing the feasibility in engineering the output voltage of Na-ion full cells.

Figure 4d presents rate performance of the Na-ion full cells with 2 M $G_1$- or 1 M $G_4$-based electrolyte at current densities from 0.05, 0.1, 0.2, 0.5, 1, and 2 to 4 A g$^{-1}$. Approximately 80% of the capacity still could be retained even when the current density increased by 80-folds from 0.05 to 4 A g$^{-1}$ for the case of the 2 M $G_1$-based electrolyte. The high rate capability renders a remarkably high-power density of 3863 W g$^{-1}$ with an energy density of 112 Wh kg$^{-1}$ (based on the total mass of electrode materials). An impressive capacity retention of 93% was also obtained after 1000 cycles, with an unprecedentedly low capacity decay rate of 0.007% per cycle (Fig. 4g), which records one of the lowest among the reported Na full cells to date[25,39–47]. The Coulombic efficiencies after ten cycles are about 99.99% (inset Fig. 4g), thus supporting their suitability to serve as energy storage devices in practical applications. To estimate the standing of our graphite||$Na_{1.5}VPO_{4.8}F_{0.7}$ (anode||cathode) full cell, we summarized the electrochemical performance of recently reported full cell configurations in terms of sodium storage capacity, output voltage, cycle life, specific energy densities, and power densities in Fig. 4e, f and Supplementary Table 2. It clearly illustrates that the graphite||$Na_{1.5}VPO_{4.8}F_{0.7}$ full cells in this work outperformed all the peers, with regard to cyclability and high-power densities, which can be attributed to the fast and stable co-intercalation reactions of graphite anodes, as well as the improved voltage by using optimal electrolytes. It is worth noting that the specific energy density of current graphite-based full cell is lower than the widely studied hard carbon-based full cell, for example, 210 Wh kg$^{-1}$ for hard carbon||$Na_{0.9}[Cu_{0.22}Fe_{0.3}Mn_{0.48}]O_2$[48], due to the larger Na storage capacity in hard carbon than that in graphite with co-intercalation reaction. However, the power density of 3863 W kg$^{-1}$ for the graphite||$Na_{1.5}VPO_{4.8}F_{0.7}$ full cell far excels the ~815 W kg$^{-1}$ for hard carbon||$Na_{0.9}[Cu_{0.22}Fe_{0.3}Mn_{0.48}]O_2$ full cell. Considering the primary applications of the secondary SIBs for large-scale stationary energy storage, the low cost of raw materials, long cycle life, and high-power capability for fast charge/discharge would be essential parameters[40], making the graphite||$Na_{1.5}VPO_{4.8}F_{0.7}$ chemistry in this work one of the most appealing choices.

The electrochemical performance and safety of secondary batteries have been demonstrated to be greatly affected by the operation temperatures[49,50]. To this end, we further evaluated the temperature dependency of cyclic capacities and rate capability of optimal graphite||$Na_{1.5}VPO_{4.8}F_{0.7}$ full cells at various temperatures. The reversible capacities at 0.1 A g$^{-1}$ were determined to be 109, 120, 121, 123, 122, and 112 mAh g$^{-1}$ at 0, 10, 25, 35, 45, and 60 °C (Supplementary Fig. 14), respectively, rendering exceptionally low-capacity variations. It is noted that the low-temperature performance of the full cell as low as 0 °C is remarkable, which is hardly achievable with conventional battery chemistry[51], possibly due to the fast reaction kinetics of co-intercalation[22] and the

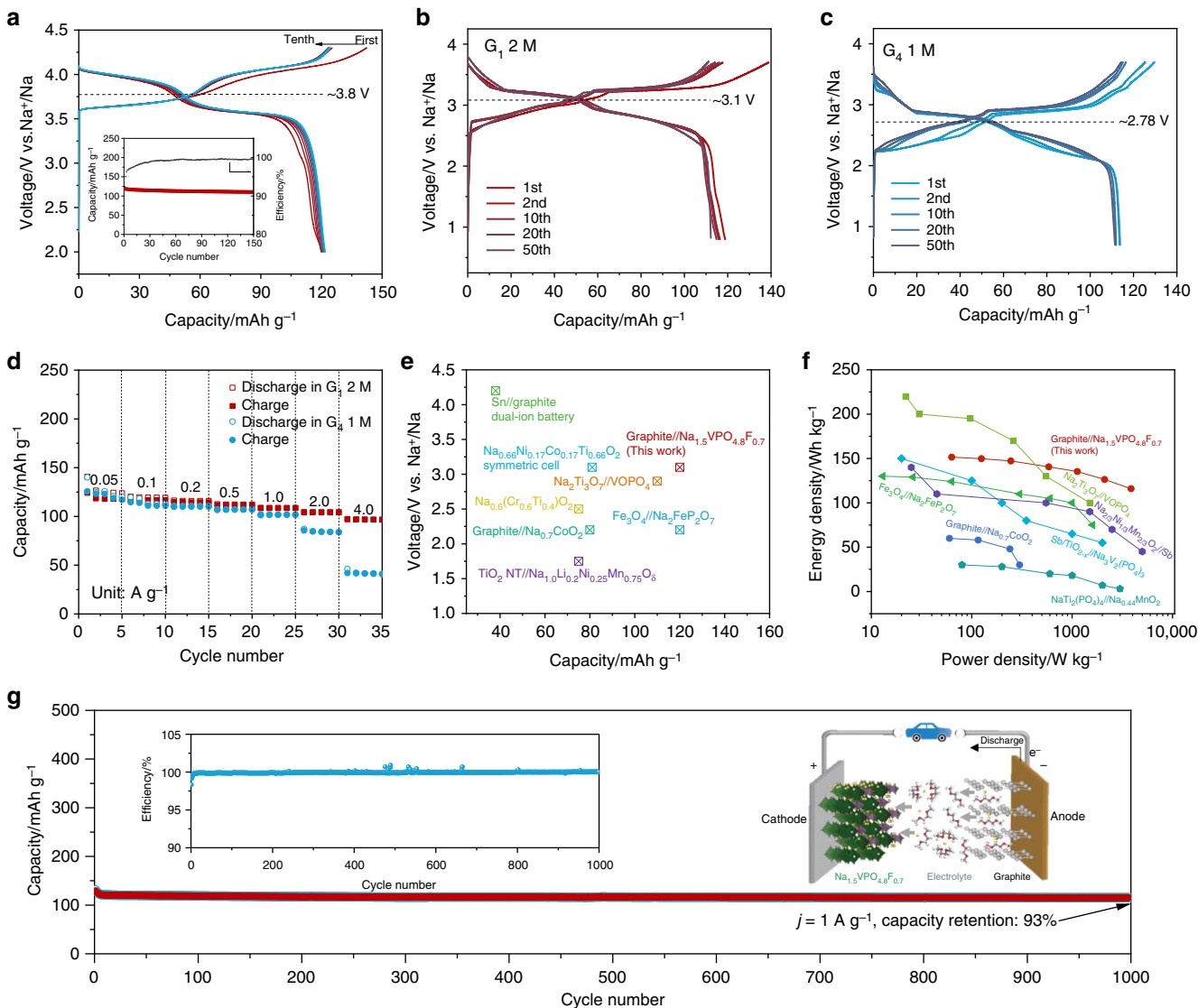

**Fig. 4** Electrochemical performance of Na ion full cells. **a** First ten cycles discharge/charge profiles, and the cyclic capacities, Coulombic efficiencies for $Na_{1.5}VPO_{4.8}F_{0.7}$ cathodes for 150 cycles in 2 M NaPF$_6$ G$_1$ electrolyte at 100 mA g$^{-1}$. **b** and **c** The 1st, 2nd, 10th, 20th, and 50th charge/discharge profiles of Na ion full cells in 2 M NaPF$_6$ G$_1$ and 1 M NaPF$_6$ G$_4$ electrolytes. **d** Rate performance of Na ion full cells in 2 M NaPF$_6$ G$_1$ and 1 M NaPF$_6$ G$_4$ electrolytes. **e**, **f** Comparison of current Na ion full cells with those recently reported in literatures[25,39–47]. **g** Long-term cycling performance of Na ion full cells at 1 A g$^{-1}$ for 1000 cycles. The schematic inset (**g**) illustrates the kinetics of solvated Na ions between two electrodes during cycling

beneficial properties from the very thin and stable SEI layer on graphite surface (Supplementary Fig. 15)[52]. In addition, the cyclic capacities measured from 0.05 to 4 A g$^{-1}$ at different temperatures (Fig. 5a) present a similarly outstanding high rate capability, while some noticeable capacity loss was observed for the case at 0 °C above 1 A g$^{-1}$. The energy and power densities of the full cells cycled at different temperatures were further evaluated, based on the total mass of the anode and cathode materials in the Ragone plot (Fig. 5b). It shows that, even under the harsh conditions of a low temperature below 10 °C, respectably high-power densities of above 1000 W kg$^{-1}$ could be successfully maintained. The outstanding temperature-dependent performance of this system would be one of the practical merits for the large-scale applications that may be exposed to the frequent change of the temperature around the year.

We have successfully developed high-energy and high-power SIBs with excellent cyclic and temperature-dependent performance by systematically investigating the systems for co-

intercalation reactions. The unique solvent-involved co-intercalation reactions require an extra solvent, potentially increasing the electrolyte/electrode mass ratio in real batteries. The ultimate electrolyte/graphite mass ratio was calculated to be 0.55 for 2 M G$_1$-based electrolyte, with a theoretical energy density of 146 Wh kg$^{-1}$, based on the total mass of anode, cathode, and the least amount of electrolyte (see detailed calculation in Supplementary Note 2). Indeed, more efforts are needed to optimize the electrolyte/electrode ratio to approach the high theoretical energy density. In addition, the optimization of a highly concentrated electrolyte would be necessary in the future for commercialization of SIBs, considering that the highly concentrated electrolyte may induce high cost. One promising attempt to mitigate this issue is to use cheap sodium salt, such as NaClO$_4$ (Supplementary Fig. 16). Another issue that needs to be taken into account for practical SIBs is the large-volume variation of graphite electrodes during cycling (Supplementary Fig. 17). Potential strategies, such as increasing the anode porosity, rationally limiting the co-

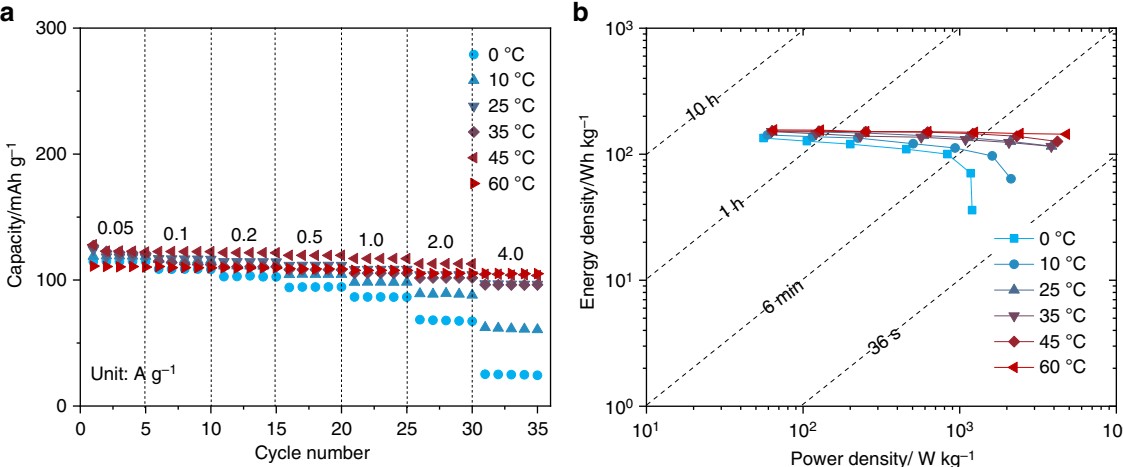

**Fig. 5** High energy and power Na-ion full cells operating at a wide range of temperature. **a** Rate capacities. **b** Ragone plots of gravimetric energy densities versus power densities for graphite||$Na_{1.5}VPO_{4.8}F_{0.7}$ full cells in an optimal electrolyte at different temperatures

intercalation capacity, and designing new cell configurations[53], are suggested.

## Discussion

We successfully tailored the co-intercalation potential of graphite anodes for Na-ion batteries with high output voltages, high energy, and power densities. By systematically investigating the factors affecting the thermodynamics of co-intercalation reactions, we demonstrated that a large tuning range of 380 mV can be achievable by adjusting the parameters of the solvent species, the concentration of electrolyte, and the temperature. Na-ion full cells with graphite anodes, $Na_{1.5}VPO_{4.8}F_{0.7}$ cathodes, and an optimal ether-based electrolyte presented an outstanding electrochemical performance, with one of the highest voltages of about 3.1 V and the competitive high-energy density of 149 Wh kg$^{-1}_{\text{both electrodes}}$. Specifically, the power density, low-temperature performance, and cycle life achievable in current Na-ion full cells is among the best thus far reported for SIBs. Moreover, the temperature-dependent charge/discharge tests exhibited excellent energy/power-density retentions at a wide temperature range of 0–60 °C. These findings support that the Na ion full cells with graphite anodes exploiting the co-intercalation are promising for large-scale energy storage system applications.

## Methods

**Preparation and characterization of electrolytes**. Electrolytes were prepared in an Ar-filled glove box at low moisture content (H$_2$O < 20 ppm) and oxygen content (O$_2$ < 0.5 ppm). Sodium salts and molecular sieves were dried in vacuum oven overnight at 180 °C, before dissolving in G$_1$, G$_2$, G$_3$, G$_4$, G$_{n-250}$ ($M_w \approx 250$ g mol$^{-1}$), and G$_{n-500}$ ($M_w \approx 50$ mol$^{-1}$) solvents at different concentrations in the glove box. Note that more information about G$_{n-250}$ and G$_{n-500}$ (supplied by Sigma Aldrich) is provided in Supplementary Fig. 18. All the solutions were strongly stirred at 50 °C for 8 h in a glove box, and then molecular sieves were added to remove the residual H$_2$O in electrolytes.

The structure of the electrolytes was studied by Raman (LabRAM HV Evolution, HORIBA, Japan) spectroscopy using a 521-nm laser at a scan range of 200–1000 cm$^{-1}$. FTIR (FT-IR-4200, JASCO, Japan) equipped with an attenuated total reflectance (ATR) was employed to examine the chemical structure of various electrolytes. The physiochemical properties of electrolytes were evaluated by using a viscosity meter (SV-10 viscometer, A&D Company Ltd., Japan) and a portable ionic conductivity meter (Model CON 610, Oakton, Singapore). A NMR spectrometer (Avance 600, Bruker, Germany) was used to study the $^{23}$Na NMR spectra of the electrolyte at room temperature, by using a liquid NMR tube. NMR spectra were recorded at $^{23}$Na frequency of 158.7 MHz, with an accumulation of 256 transients and a repetition time of 1.45 s for each sample (acquisition time for 0.45 s and relaxation delay for 1 s) for complete relaxation. All chemical shifts of $^{23}$Na NMR spectra were referenced to a standard solution of 1 M NaCl aqueous solution.

**Electrode preparation and electrochemical characterizations**. The graphite anodes were prepared by mixing natural graphite powders (average size ≈100 μm) and a polyvinylidene fluoride (PVDF) binder with a mass ratio of 9:1, using N-methyl-2-pyrrolidone (NMP) solvent. The uniform slurry was then cast on Cu foil and dried at 70 °C overnight in a vacuum oven. The average electrode thickness and active material loading were ~40 μm and ~3 mg cm$^{-2}$, respectively. The $Na_{1.5}VPO_{4.8}F_{0.7}$ cathode materials were synthesized according to our previous work[35]. Specifically, a stoichiometric amount of V$_2$O$_5$ (Sigma Aldrich, 99%) and NH$_4$H$_2$PO$_4$ (Sigma Aldrich, 99%) were blended by ball milling and annealing at 750 °C for 4 h in Ar atmosphere to obtain VOPO$_4$ powder. To synthesize VPO$_4$ powder, a stoichiometric amount of V$_2$O$_5$ and NH$_4$H$_2$PO$_4$ was mixed together and ball milled with 20 mol% super P, and then heat-treated at 850 °C for 2 h in Ar atmosphere. VOPO$_4$, VPO$_4$, NaF (Sigma Aldrich, 99%), and Na$_2$CO$_3$ (Sigma Aldrich, 99%) precursors were later mixed with a molar ratio of 8:2:7:4. Blending of the precursors was conducted by high-energy ball milling at 300 rpm for 24 h; the resulting mixture was heat-treated at 750 °C for 1.5 h under flowing Ar. The $Na_{1.5}VPO_{4.8}F_{0.7}$ cathodes were fabricated by mixing 80 wt% $Na_{1.5}VPO_{4.8}F_{0.7}$ powders, 10 wt% super P, and 10 wt% PVDF binder in NMP solvent. The loadings of the active materials on the electrodes were ~3 mg cm$^{-2}$ for half-cell tests.

Sodium ion half-cells were assembled in a glove box into a two-electrode configuration, using the working electrodes, sodium metal as the counter electrode, glass fiber (GF/F, Whatman, USA) as a separator, and 100-μl electrolyte. Graphite|| $Na_{1.5}VPO_{4.8}F_{0.7}$ full cells were fabricated by using graphite anodes and $Na_{1.5}VPO_{4.8}F_{0.7}$ cathodes, with an optimal mass ratio of 1:1.5 of active materials. For full cell measurement, the loadings of the graphite anode were about 6 mg cm$^{-2}$, on the same scale with the literature[48]. The cathode pellet is slightly larger than the anode (2/3 inch for cathode and 1/2 inch for anode, respectively). The energy and power densities for full cells were calculated based on the total mass of active materials, excluding masses of other components such as separator, electrolyte, and coin cell components[42–50]. It is noted that pre-sodiation for anodes was performed to activate the material and stabilize the electrode surface. The electrochemical performance of as-prepared coin cells was measured on a multichannel potentio-galvanostat (Won A Tech, Korea) at different current densities. The voltage windows for half-cells with graphite anodes, half-cells with $Na_{1.5}VPO_{4.8}F_{0.7}$ cathodes, and the full cells were set as 0.1 –2.0 V, 2.0–4.3 V, and 0.7–3.8 V, respectively.

**Calculations**. To estimate the repulsion energy between negatively charged graphene layers, density functional theory calculations were conducted. It is noted that the amount of charge injected in graphene layers was also properly determined, based on the reversible capacity of different GICs, similar to our previous work[27]. See Supplementary Information for detailed computational methods.

## Data availability

The data that support the findings of this study are in the paper and/or the Supplementary Information. Additional data are available from the authors on reasonable request.

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

## Acknowledgements

This work was financially supported by Samsung Research Funding Center of Samsung Electronics under project No. SRFC-TA1603-03 and the Creative Materials Discovery Program through the National Research Foundation (NRF) under project No. NRF-2017M3D1A1039553. Z.L.X. acknowledges the Korea Research Fellowship (KRF) Program through the National Research Foundation of Korea funded by the Ministry of Science and ICT (Project No. 2017H1D3A1A01013931).

## Author contributions

K.K. and Z.L.X. conceived the original idea. Z.L.X. designed the experiments with help from K.Y.P., H.P., O.T., S.J.K. and W.M.S. The computations were performed by G.Y. K. K. revised the paper prepared by Z.L.X. All authors discussed the results and contributed to the final manuscript.

## Additional information

**Competing interests:** The authors declare no competing interests.

