## [Peer Review File · Nature Communications]

Reviewers' comments:

Reviewer #1 (Remarks to the Author):

This is an interesting study. I think the readers of Nature Communications may find it of value. Overall, the work here is innovative, and the article is written well. However, there are some critical issues the authors should carefully address as follows.

The manuscript criticizes the rate capability of hard carbon. This needs to be corrected as it has been shown that the seemingly low rate capability was all due to the 2-electrode cell configuration, which dwarfs the kinetics of hard carbon. If 3-electrode cells are used without stressing the reference electrode, the rate behavior of hard carbon could indeed be excellent.

What is the saturation concentration for the electrolyte salts, the best one used here? Why not explore the limits of the benefits afforded by having an ever-higher concentration? What is the sweet spot if an even high one is unfavored for some reason?

The energy density calculation of the full cell based on the anode and cathode seems insufficient to truly represent the unique chemistry for the cell that employs the anode utilizing co-intercalation. The mass of the solvent should be considered due to its indispensability for the chemistry here. With the necessity to have some extra solvent in the electrolyte in this chemistry, what would be the ultimate electrolyte/electrode mass ratio looking like?

A fundamental challenge of any batteries that utilize a co-intercalation graphite anode is that such an anode swells to be big when converting to its ternary compounds.

What is the volume expansion by how many folds in the case here with the highest capacity recorded?

In a practical cell, how to combat this volume change in the full-cell design?

What is the active mass of loading of the electrodes of the anode here? When giving a comparison of 3863 W/kg to 815 W/kg for the power, where the latter belongs to the hard-carbon-based full cell, did the authors compare the active mass loading? Are they on the same scale?

When the operation potential is lowered to this new level, is the SEI layer still absent? Any evidence to support such a claim?

There has been minimal support from computational studies here to buttress the theoretical claims. By lacking calculations, I feel that the theory here, albeit soundly empirical, is relatively lean in its strengths and applicability.

Reviewer #2 (Remarks to the Author):

Summary: The presented manuscript systematically studies the electrochemical formation of ternary graphite intercalation compounds (t-GICs) upon the co-intercalation of Na-ions solvated by different ether-based solvent molecules. Different parameters, such as the ether length, the electrolyte concentration and temperature affect the co-intercalation potential, which can be exploited to increase the voltage of Na-ion batteries. After very careful half-cell studies on each parameter, the optimal electrolyte was chosen to construct and operate a graphite|Na_{1.5}VOP_{4.8}F_{0.7} full cell Na-ion battery, which shows a high cycling stability and an excellent performance for fast charging/discharging. Both

properties are strongly required for stationary large scale batteries.

Evaluation: The paper is very well written and does not leave many open questions. The language is very clear, free of mistakes and easy to follow. The observed data and their interpretation seem mostly reasonable and new. This manuscript fits into the scope of Nature Communications and might reach the impact of this journal. Before acceptance, some minor issues should be addressed:

- Line 45: '...', forming ternary ...': insert an article as '...', forming a ternary ...'
- Line 55: The phrase 'without a significant solid electrolyte interphase (SEI) layer' is misleading. Maibach et al. for example dedicated a full study on the SEI layer in these systems (J. Maibach, F. Jeschull, D. Brandell, K. Edström, and M. Valvo, ACS Appl. Mater. Interfaces, 9, 12373–12381 (2017)). There also was some SEI-formation visible in reference 22. Indeed, the SEI layer thickness does not seem to exceed 10 nm, however an SEI-formation still takes place. I am also aware of reference 52, where the title hints to a missing SEI layer. Reference 52, however, also finds indications for an SEI-formation, though not comparable to the SEI in Li-ion batteries. Glymes are also often employed as electrolyte additive to form beneficial SEI-properties. Hence, my suggestion is to rephrase the sentence as follows: '... with the growth of a thin solid electrolyte interphase (SEI) layer.' It seems, as the SEI-formation in the studied system still requires more attention by the research community.
- Line 99: What exactly is the Gn-250 and Gn-500 solvent? Considering the molar weight of Gn-250, it seems as if it was G5? Do the authors have any idea, how many Gn-250 molecules solvate one Na-ion? Can Gn-500 be considered as G10 or G11? Are the Gn-250 and Gn-500 linear glymes at all? It would be highly interesting to see, how the Na-ion is solvated by these molecules. Is there any further characterization of these solvents? Are these solvents self-synthesized? The Methods part does not address, where and in which purity the solvents were purchased or whether they were self-synthesized. Can the authors please add the missing information?
- Line 107: How was the 'average voltage' determined? From Fig 1b it seems that the 'average Na co-intercalation voltage' coincides with the onset potential of the sharp intercalation peak, at least for the G1, G2 and G4 electrolytes. For the Gn-250 and Gn-500 this, however, does not seem to be the case. A good way to determine the 'average voltage' is to determine the energy density E by integrating the voltage-profile over the specific capacity Q . The average voltage V then can be calculated according to $V=E/Q$. Can the authors please comment on the determination of the 'average voltage' and whether it is just coincidence that the average voltage appears to be the onset potential of the sharp reduction peak in the G1, G2 and G4 electrolytes?
- Line 112 and following: It is discussed that the Na-ions are more efficiently shielded by longer solvents, which weakens the interaction with the graphite lattice. For my own curiosity: Is there any experience, whether the rate capability improves with longer ether solvents?
- Supplementary Fig 7: Why the reversible capacity of graphite with the 3 M LiPF₆ EC/DEC electrolyte at 25°C is such low? Is this result reproducible? Is there any explanation?
- Line 270 and Fig 4a, g: The scale bars of the coulombic efficiencies are not very useful, as one cannot see, how close the coulombic efficiency is to 100%. Can the authors please magnify the scale, maybe from 90% to 105%? It would also be helpful for the reader to mention an exact number of the coulombic efficiency in the text after e.g. 100 cycles instead of 'close to 100%'.
- Line 296: I do not fully agree with 'the absent SEI layer on graphite surfaces'. As discussed above, this question must be addressed by more detailed studies, which are not the scope of this manuscript. Certainly, the SEI film must be very thin, sustain large volume changes and have a minor influence on the co-intercalation kinetics, but I would not call it 'absent SEI layer'. A more vague wording such as '... and the beneficial properties of the thin SEI layer...' might solve this issue.
- Line 332: Please add the units for the molar masses...
- General: Sometimes the unit after a value is separated by a space, sometimes not. Please use a consistent style.
- Typo in Supplementary Table 1: Unit of the energy density should be '2.5 Wh kg⁻¹' instead of '2.5 Wk kg⁻¹' in the line of reference 7.

One major issue should also be addressed before acceptance:

- Fig 2 and Supplementary Fig 5: The concentration dependent potential shift of the co-intercalation potential might also simply be caused due to the change of the reference potential of the Na-electrode. A way, how one could experimentally rule out, whether the observed effect is an artefact due to a flawed reference electrode is to add a certain amount of ferrocene to the electrolyte and use the ferrocene peak as a reference. I would suggest to perform this experiment for the 0.05 M, the 1 M and the 3 M electrolytes.

Authors' response to Reviewers' Comments:

The authors sincerely appreciate the invaluable comments offered by the reviewers and the editor. All comments are now incorporated in the revision and the summary is presented in the following of how the amendments are made according to the individual comment.

Reviewer #1 (Remarks to the Author):

This is an interesting study. I think the readers of Nature Communications may find it of values. Overall, the work here is innovative, and the article is written well. However, there are some critical issues the authors should carefully address as follows.

Reply: We are excited that the reviewer estimated our work to be interesting to the readers of Nature Communications. We are also beyond grateful to the reviewer for raising constructive comments/suggestions to improve the quality of our paper.

1. The manuscript criticizes the rate capability of hard carbon. This needs to be corrected as it has been shown that the seemingly low rate capability was all due to the 2-electrode cell configuration, which dwarfs the kinetics of hard carbon. If 3-electrode cells are used without stressing the reference electrode, the rate behavior of hard carbon could indeed be excellent.

Reply: We are grateful to the reviewer for the valuable suggestion. The following sentence in Line 36 is corrected accordingly:

~~“Regarding the use of the hard carbon anode, a poor rate capability is the challenging issue to be resolved, and the low redox voltage (0.1-0 V vs. Na⁺/Na) close to that of the sodium metal plating would induce serious concerns on the safety issues.¹²”~~

2. What is the saturation concentration for the electrolyte salts, the best one used here? Why not explore the limits of the benefits afforded by having an ever-higher concentration? What is the sweet spot if an even high one is unfavored for some reason?

Reply: We appreciate this constructive suggestion from the reviewer. As discussed in the original manuscript, we claimed that the co-intercalation voltage would downshift with increasing the concentration of electrolyte. Highly concentrated 2 M G₁-based electrolyte was favorable for low co-intercalation potential. According to the reviewer's comment, we additionally prepared G₁-

based electrolytes with higher concentrations, *i.e.*, 2.5 M and 3 M. However, when the electrolyte concentration increased from 2 M to 3 M, the viscosity dramatically surged from of 3.1 cP to 12.7 cP, and the ionic conductivity decreased from 16.6 mS to 9.1 mS. The charge transfer resistance also largely increased for electrodes in 3 M electrolyte (Supplementary Fig. 8). As a result, the rate capabilities of graphite cycled in 3 M electrolyte are much inferior to those in 2 M. Therefore, it is believed that 2 M G₁-based electrolyte was close to the sweep spot for optimal battery performance in this work. The saturation concentration was identified as 3 M for G₁-based electrolyte. We observed large amounts of salt precipitates in the electrolyte if we further increase the electrolyte concentration to above 3 M. The experimental result can be explained by evaluating the Na ion coordination property in G₁ electrolyte. It was reported that the O coordination number of Na ion with ether functional groups was suggested to be in the range of 4 to 6, and considering the two C-O-C bonds in one G₁ molecule, the molar ratio between G₁ and Na ion for fully coordinated solution is estimated to be 3 (*Energy Environ. Sci.* 2017, 10(7), 1631-1642). In 3 M G₁-based electrolyte, the solvent-to-salt molar ratio is calculated to be 3.2. Thus, it is reasonable to consider 3 M as the highest achievable concentration for G₁-based electrolyte at room temperature.

As a response to the reviewer's valuable comments, the following sentences are added in the manuscript:

“It is worth noting that we also prepared G₁-based electrolyte with higher concentrations (*i.e.*, 2.5 M and 3 M) to explore the limit of using highly concentrated electrolyte to further decrease co-intercalation voltage. However, electrodes cycled in 2.5 M and 3 M G₁-based electrolytes presented inferior rate capabilities (Supplementary Fig. 8), due to the high viscosity of the electrolytes. 2 M G₁-based electrolyte was thus identified as close to the sweet spot for further electrochemical evolutions.”

The following figure and discussion are added in Supplementary Information:

Supplementary Figure 8 (a) Discharge/charge profiles of graphite electrodes in 2 M, 2.5 M and 3 M G₁-based electrolytes at 0.1, 0.2, 0.5, 1 and 2 A g⁻¹, respectively, (b) Nyquist plots of cycled electrodes in (a), (c) ionic conductivity and (d) viscosity of G₁-based electrolytes with concentrations ranging from 0.05 M to 3 M. According to the discussion in main context, higher electrolyte concentrations lead to lower co-intercalation voltages. To explore the limit of the benefits afforded from highly concentrated G₁-based electrolyte, we prepared electrolytes to 3 M, which is the saturation concentration at room temperature. In (a), it shows that the rate capacities decrease dramatically by using 2.5 M and 3 M G₁-based electrolytes. The poor battery performance is attributed to the (b) high charge transfer resistance, (c) the largely decreased ionic conductivity and (d) the significantly increased viscosity for 2.5 M and 3 M G₁-based electrolytes. In contrast, 2 M G₁-based electrolyte possesses the highest ionic conductivity and a moderate viscosity. Therefore, 2 M G₁-based electrolyte is considered close to the sweet spot in this work.

3. The energy density calculation of the full cell based on the anode and cathode seems insufficient to truly represent the unique chemistry for the cell that employs the anode utilizing co-intercalation. The mass of the solvent should be considered due to its indispensability for the

chemistry here. With the necessity to have some extra solvent in the electrolyte in this chemistry, what would be the ultimate electrolyte/electrode mass ratio looking like?

Reply: We appreciate the reviewer for raising this important question. The optimal full cells in this work use 2 M NaPF₆ G₁ electrolyte, graphite anodes and Na_{1.5}VPO_{4.8}F_{0.7} cathodes. The sodium storage capacity in graphite is about 120 mAh g⁻¹, leading to a [Na-G₁]C₁₉ ternary graphite intercalation compound (t-GIC). Based on our previous work (*Energy Environ. Sci.* 2015, 8, 2963), the molar ratio of Na ion: G₁ of intercalated solvated-Na-ion is 1. The Na ions and G₁ molecules are provided by Na_{1.5}VPO_{4.8}F_{0.7} cathode and electrolyte, respectively. Thus, when we assume that 1 mol graphite is fully co-intercalated with the solvated-Na-ion, the theoretical content of G₁ molecules from electrolyte should be 1/19 mol. Then, the least amount of electrolyte is calculated to be 6.58 g (in 2 M NaPF₆ DME electrolyte, the molar ratio of NaPF₆: DME is 2: 9.62; thus the mass of electrolyte can be calculated as $\frac{1}{19} mol \times 90.12 \frac{g}{mol} + \frac{1}{19} mol \times \frac{2}{9.62} \times 167.9 \frac{g}{mol} = 6.58g$). The ultimate electrolyte/graphite electrode mass ratio is evaluated to be 0.55 (or 0.22 for electrolyte/electrodes with cathode: anode mass ratio = 1.5 in this work). Given the theoretical capacities of about 120 mAh g⁻¹ for graphite anode (Fig. 3g) and Na_{1.5}VPO_{4.8}F_{0.7} cathode (Fig. 4a) and an output voltage of 3.1 V, the theoretical energy density of a graphite||Na_{1.5}VPO_{4.8}F_{0.7} full cell is calculated to be 146 Wh kg⁻¹ (*i.e.*, $120 \times 3.1 \times \frac{12}{12+12+6.58}$), based on the total mass of anode, cathode and the least amount of electrolyte. In this work, an optimal full cell possesses a cathode: anode mass ratio of 1.5, a graphite mass loading of 6 mg cm⁻² (electrode diameter = ½ inch) and 100 μl electrolyte (*i.e.*, mass density of about 1.0 mg μl⁻¹), thus the energy density is calculated to be 23.8 Wh kg⁻¹ (*i.e.*, $120 \times 3.1 \times \frac{7.6}{7.6+7.6 \times 1.5+100}$). Apparently, the practical energy density is far from the theoretical value, due to the flooded amount of electrolyte in this work. More efforts are ongoing to decrease the electrolyte/electrode mass ratio to approach the high theoretical energy density of graphite||Na_{1.5}VPO_{4.8}F_{0.7} full cells.

In response with the reviewer's comment, the following sentences are added in the revised manuscript:

“We have successfully developed high energy and power SIBs with excellent cyclic and temperature-dependent performance by systematically investigating the systems for co-intercalation reactions. The unique solvent involved co-intercalation reactions require extra solvent, potentially increasing the electrolyte/electrode mass ratio in real batteries. The ultimate electrolyte/graphite mass ratio was calculated to be 0.55 for 2 M G₁-based electrolyte, with a theoretical energy density of 146 Wh kg⁻¹ based on the total mass of anode, cathode and the least amount of electrolyte (see detailed calculation in Supplementary Note 2). Indeed, more efforts are needed to optimize the electrolyte/electrode ratio to approach the high energy density.”

The following paragraph is added Supplementary Information:

“**Supplementary Note 2** | Energy density of optimal graphite||Na_{1.5}VPO_{4.8}F_{0.7} full cells

The optimal full cells in this work use 2 M NaPF₆ G₁ electrolyte, graphite anodes and Na_{1.5}VPO_{4.8}F_{0.7} cathodes. The sodium storage capacity in graphite is 120 mAh g⁻¹, leading to a [Na-G₁]C₁₉ ternary graphite intercalation compound (t-GIC). Based on our previous work, the molar ratio of Na ion: G₁ of intercalated solvated-Na-ion is 1.²³ The Na ions and G₁ molecules are provided by Na_{1.5}VPO_{4.8}F_{0.7} cathode and electrolyte, respectively. Thus, when we assume that 1 mol graphite is fully co-intercalated with the solvent, the theoretical content of G₁ molecules from electrolyte should be no less than 1/19 mol. Then, the least amount of electrolyte is calculated to be 6.58 g (in 2 M NaPF₆ DME electrolyte, the molar ratio of NaPF₆: DME is 2: 9.63; thus the mass of electrolyte can be calculated as $\frac{1}{19} \text{ mol} \times 90.12 \frac{\text{g}}{\text{mol}} + \frac{1}{19} \text{ mol} \times \frac{2}{9.63} \times 167.9 \frac{\text{g}}{\text{mol}} = 6.58 \text{ g}$). The ultimate electrolyte/graphite electrode mass ratio is evaluated to be 0.55 (or 0.22 for electrolyte/electrodes with cathode: anode mass ratio =1.5 in this work). Given the theoretical capacities of about 120 mAh g⁻¹ for graphite anode (Fig. 3g) and Na_{1.5}VPO_{4.8}F_{0.7} cathode (Fig. 4a) and an output voltage of 3.1 V, the theoretical energy density of a graphite||Na_{1.5}VPO_{4.8}F_{0.7} full cell is calculated to be about 146 Wh kg⁻¹ based on the total mass of anode, cathode and the least amount of electrolyte. In this work, we used excess amount of electrolyte (*i.e.*, 100 μl , mass density of about 1 mg μl^{-1}) to estimate the cyclic stability of graphite||Na_{1.5}VPO_{4.8}F_{0.7} full cells. The electrolyte/electrode mass ratio is calculated to be 5.26, considering the anode: cathode mass ratio = 1.5 and graphite mass loading of 6 mg cm⁻² (electrode diameter = 1/2 inch). Accordingly, the practical energy density is estimated to be only

23.8 Wh kg⁻¹, which is far from the theoretical value. Apparently, more efforts are needed to decrease the electrolyte/electrode ratio to approach the high theoretical energy density.”

23. Kim, H. *et al.* Sodium intercalation chemistry in graphite. *Energy Environ. Sci.* **8**, 2963–2969 (2015).

4. A fundamental challenge of any batteries that utilize a co-intercalation graphite anode is that such an anode swells to be big when converting to its ternary compounds. What is the volume expansion by how many folds in the case here with the highest capacity recorded? In a practical cell, how to combat this volume change in the full-cell design?

Reply: We are grateful to the reviewer for raising the important questions. We conducted cross-sectional SEM images of pristine, 1st discharged and 10th discharged anodes to show the volume expansion of fully co-intercalated graphite in this work (Supplementary Fig. 15). It reveals that the volume expansion is near 80 %. The graphite electrodes were reported to periodically expand/contract by about 70-100 % during cycling based on an *in-situ* electrochemical dilatometry study (*Adv. Energy Mater.* 2018, 1702724). Generally, the large volume expansion would induce instability of active materials and poor cyclic stability of electrodes, such as silicon anodes in Li-ion batteries. However, it was found that the large volume expansion does not injure the long cyclic life of graphite electrodes, due to the van der Waals interaction between graphene layers and solvated-Na-ion (*Nano Energy* 2017, 34, 456-462). For Na-ion full cells, the volume deformation should be strictly limited for electric device applications. Usually the volume expansion of commercial Li-ion battery at cell level should be no more than 20 %. In view of the cavities in electrodes, the gap between the electrode and the separator as well as the cushion from the separator, volume variation for anode materials should be limited to 70 % (*J. Electrochem. Soc.* 2015, 162 (14) A2509-A2528). The volume expansion of fully co-intercalated graphite anode is marginally larger than the requirement for commercial Li-ion batteries. To accommodate this issue, several potential strategies are suggested: (i) to increase the porosity of graphite anodes, *e.g.*, using graphite foam electrodes (*Nano Lett.* 2016, 16, 543-548), (ii) to rationally limit the co-intercalation capacity, and (iii) to design new cell configurations.

In response with the reviewer’s valuable comments, the following sentences and reference are added in the revised manuscript:

“Another issue that needs to be taken into account for practical SIBs is the large volume variation of graphite electrodes during cycling (Supplementary Fig. 17). Potential strategies, such as increase the anode porosity, rationally limit the co-intercalation capacity and design new cell configurations⁵³, are suggested.”

53. Luo, F. *et al.* Nanosilicon/carbon composite anode materials towards practical application for next generation Li-ion batteries. *J. Electrochem. Soc.*, **162** (14) A2509-A2528 (2015).

The following SEM images are added in Supplementary Information:

Supplementary Figure 17 SEM images of (a) pristine, (b) 1st discharged, and (c) 10th discharged graphite electrodes. The volume expansion is calculated to be about 80 % and remains stable during cycles.

5. What is the active mass of loading of the electrodes of the anode here? When giving a comparison of 3863 W/kg to 815 W/kg for the power, where the latter belongs to the hard-carbon-based full cell, did the authors compare the active mass loading? Are they on the same scale?

Reply: We need to thank the reviewer for the careful review. When giving a comparison of 3863 W kg⁻¹ for our graphite//Na_{1.5}VPO_{4.8}F_{0.7} to 815 W kg⁻¹ for hard carbon//Na_{0.9}[Cu_{0.22}Fe_{0.3}Mn_{0.48}]O₂, we used a mass loading of about 6 mg cm⁻² for graphite anode. This value is on the same scale with the 6 to 10 mg cm⁻² for hard carbon in the hard carbon//Na_{0.9}[Cu_{0.22}Fe_{0.3}Mn_{0.48}]O₂ full cell.

The following information is added in Methods:

“For full cell measurement, the loadings of the graphite anode were about 6 mg cm⁻², on the same scale with the literature.⁴⁸”

6. When the operation potential is lowered to this new level, is the SEI layer still absent? Any evidence to support such a claim?

Reply: We agree with the reviewer that it is a good idea to check the SEI layer on graphite cycled under optimal conditions. The TEM image and XPS result are added in Supplementary Fig. 15. The following sentence in manuscript is also modified accordingly:

“possibly due to the fast reaction kinetics of co-intercalation²² and the beneficial properties of the very thin and stable SEI layer on graphite surface (Supplementary Fig. 15)⁵².”

The following figure is added in Supplementary Information:

Supplementary Figure 15 TEM and XPS analyses of SEI layer on graphite electrodes after 5 cycles in 2 M NaPF₆ G₁ electrolyte. TEM image indicates that no noticeable SEI layer was formed at the surface of graphite. The XPS results with depth profiling (from surface to 20 nm) also clearly show that the amount of SEI on the surface of cycled graphite is rather trivial.

7. There has been minimal support from computational studies here to buttress the theoretical claims. By lacking calculations, I feel that the theory here, albeit soundly empirical, is relatively lean in its strengths and applicability.

Reply: We appreciate the reviewer's valuable comment on the role of calculation results. As discussed in the manuscript, we claimed that the co-intercalation voltage depends on (i) the solvent species, (ii) the activity of solvents and (iii) the temperature. Since the origins of factors (ii) and (iii) can be clearly demonstrated by experiments or classical thermodynamics, we focused our computational efforts on revealing the origin of factor (i). As a result, with the aid of DFT calculations, we successfully analyzed the relative stability of t-GICs. Nevertheless, we agree to the reviewer's comment that the additional calculations would help strengthen our theory. In this regard, we additionally performed DFT calculations to compare the amount of charge transferred from intercalants to graphene layers for a series of t-GICs, which would give

us critical information on the relative stability of t-GICs as it determines the magnitude of repulsive interaction between negatively charged graphene layers. Our Bader charge analysis combined with the information on the reversible capacities in Supplementary Fig. 1 revealed that the amount of charge transferred from intercalants to graphene layers is almost identical for G₁, G₂, G₄ co-intercalated t-GICs. This result allows us to compare the repulsive interactions of all compounds in a single curve in Fig. 1c without the assumption of the identical charge transfer, which strengthens our theory on the relative stability of t-GICs.

In response with the reviewer’s comment, the following sentences are added in the revised manuscript:

“When the relative stabilities of the graphite host are estimated as a function of interlayer distance from the DFT calculations as depicted in Fig. 1c and Supplementary Fig. 2, it is observed that the repulsion energy decreases with the distance between graphene layers. We would like to note that the amount of charge transferred from the intercalants to graphene layers is almost identical among co-intercalated t-GICs, because of their similar reversible capacities (Supplementary Fig. 1) and Bader charges of intercalants (Supplementary Table 1). This allows us to compare the repulsive interactions of all compounds in a single curve in Fig. 1c.”

The calculation method, result and reference are added in Supplementary Information:

“In order to obtain the amount of charge transferred from intercalants to graphene layers, Bader charge analysis was conducted for G₁, G₂ and G₄ co-intercalated graphite.^{22,}”

Supplementary Table 1 Bader charge of [Na-G_n]⁺ complex in G₁, G₂, G₄ co-intercalated t-GICs. Regardless of the solvent species, amount of charge transfer is almost identical.

[Na-G _n] ⁺ complex	[Na-G ₁] ⁺	[Na-G ₂] ⁺	[Na-G ₄] ⁺
Bader charge	+0.89	+0.89	+0.90

22. Tang, W., Sanville, E., Henkelman, G. A grid-based Bader analysis algorithm without lattice bias, *J. Phys.: Condens. Matter* **21**, 084204 (2009).

Reviewer #2 (Remarks to the Author):

Summary: The presented manuscript systematically studies the electrochemical formation of ternary graphite intercalation compounds (t-GICs) upon the co-intercalation of Na-ions solvated by different ether-based solvent molecules. Different parameters, such as the ether length, the electrolyte concentration and temperature affect the co-intercalation potential, which can be exploited to increase the voltage of Na-ion batteries. After very careful half-cell studies on each parameter, the optimal electrolyte was chosen to construct and operate a graphite||Na_{1.5}VOP_{4.8}F_{0.7} full cell Na-ion battery, which shows a high cycling stability and an excellent performance for fast charging/discharging. Both properties are strongly required for stationary large scale batteries.

Evaluation: The paper is very well written and does not leave many open questions. The language is very clear, free of mistakes and easy to follow. The observed data and their interpretation seem mostly reasonable and new. This manuscript fits into the scope of Nature Communications and might reach the impact of this journal. Before acceptance, some minor issues should be addressed:

Reply: We would like to thank the reviewer for the positive comments and highlight on our findings about co-intercalation reactions. In addition, the valuable suggestions to further improve the quality of our manuscript are highly appreciated.

1. • Line 45: ‘..., forming ternary ...’: insert an article as ‘..., forming a ternary ...’

Reply: The below sentence in Line 45 is corrected:

“solvated-sodium-ions are intercalated into the galleries of graphite, forming a ternary graphite intercalation compound.”

2. • Line 55: The phrase ‘without a significant solid electrolyte interphase (SEI) layer’ is misleading. Maibach et al. for example dedicated a full study on the SEI layer in these systems (J. Maibach, F. Jeschull, D. Brandell, K. Edström, and M. Valvo, ACS Appl. Mater. Interfaces, 9, 12373–12381 (2017)). There also was some SEI-formation visible in reference 22. Indeed, the SEI layer thickness does not seem to exceed 10 nm, however an SEI-formation still takes place. I am also aware of reference 52, where the title hints to a missing SEI layer. Reference 52, however, also finds indications for an SEI-formation, though not comparable to the SEI in Li-ion batteries. Glymes are also often employed as electrolyte additive to form beneficial SEI-

properties. Hence, my suggestion is to rephrase the sentence as follows: ‘... with the growth of a thin solid electrolyte interphase (SEI) layer.’ It seems, as the SEI-formation in the studied system still requires more attention by the research community.

Reply: We are grateful to the reviewer for the insightful comment. We agree to the reviewer’s comment to rephrase the below sentence in Line 55:

“It was also unveiled that the co-intercalation in graphite occurs via a fast staging process^{21,22} with the growth of a very thin solid electrolyte interphase (SEI) layer.”

3. • Line 99: What exactly is the G_n-250 and G_n-500 solvent? Considering the molar weight of G_n-250, it seems as if it was G5? Do the authors have any idea, how many G_n-250 molecules solvate one Na-ion? Can G_n-500 be considered as G10 or G11? Are the G_n-250 and G_n-500 linear glymes at all? It would be highly interesting to see, how the Na-ion is solvated by these molecules. Is there any further characterization of these solvents? Are these solvents self-synthesized? The Methods part does not address, where and in which purity the solvents were purchased or whether they were self-synthesized. Can the authors please add the missing information?

Reply: We need to thank the reviewer for raising these interesting questions. G_n-250 and G_n-500 are poly (ethylene glycol) dimethyl ethers with average molecular weight of about 250 and 500 g mol⁻¹, respectively. These chemicals are purchased from Sigma-Aldrich (CAS No. 24991-55-7) with purity of above 99.5%. The chemical formulas for G_n are CH₃O[OCH₂CH₂]_nCH₃, belonging to linear ethers. G_n-250 and G_n-500 are not pure ether molecules, but mixtures of long chain ethers. Generally, G_n-250 is considered as a mixture of G_n with an average $n = 4.6$, while G_n-500 is regarded as G_n with an average $n = 10.3$ (*J. Phys. Chem.* 1994, 98, 8234-8244; *J. Power Sources*, 2015, 299, 460-464). Although detailed electrolyte structures with G_n-250 and G_n-500 solvents are difficult to be identified, the co-intercalation voltages of graphite in G_n-250 and G_n-500-based electrolytes agreed well with our statement that an increase of the average co-intercalation voltage was obtained as the average chain length of the solvent increases (*i.e.*, average voltages of 0.60, 0.66, 0.77, 0.78 and 0.81 V for G_n-based electrolytes with $n = 1, 2, 4, 4.6$ and 10.3, respectively).

It was reported that for $n \geq 2$, *i.e.*, when three or more coordination sites (*i.e.*, oxygen atoms) are available in the G_n ethers, the molar ratio of G_n : Na ion is essentially 1 (*J. Am. Chem. Soc.* 1967, 89 (17) 4547; *J. Am. Chem. Soc.* 1970, 92(7) 1955). That means one G_{n-250} or G_{n-500} ether molecule tends to coordinate with one Na ion in the solvent-separated ion pair. In addition, we can quantify the number of co-intercalated G_{n-250} or G_{n-500} molecules per Na ion in the graphite by monitoring the weight change of graphite at different states of charge and discharge. It is observed that the weight change of the intercalated graphite follows the G_n : Na =1, suggesting one Na ion is solvated with about one G_{n-250} or G_{n-500} molecules in graphite intercalation compounds.

As a response to the reviewer's comments, the follow sentence is added in Methods:

“Note that more information about G_{n-250} and G_{n-500} (supplied by Sigma Aldrich) are provided in Supplementary Fig. 18.”

The following figure and discussion are added in Supplementary Information:

Supplementary Figure 18 Weight change of graphite measured in various states of sodiation and desodiation to determine the b values in (a) $[\text{Na}(\text{G}_{n-250})_b]_x \text{C}_{21}$ and (b) $[\text{Na}(\text{G}_{n-500})_b]_x \text{C}_{22}$. In both cases, b is near to be 1, suggesting one Na ion is solvated with one G_{n-250} or G_{n-500} molecule in co-intercalated graphite. It is worth noting that G_{n-250} and G_{n-500} are mixture of long chain ethers (*i.e.*, G_n with average $n = 4.6$ and 10.3 , respectively). Although they are not pure ether molecules, the co-intercalation voltages of graphite in G_{n-250} and G_{n-500} -based electrolytes also followed the trend that the average co-intercalation voltage increases as the average chain length

of the solvent increases. In addition, one G_{n-250} or G_{n-500} ether molecule tends to coordinate with one Na ion in the electrolyte according to previous studies.^{1,2}

1. Chan, L. L., and Smid, J. Contact and solvent-separated ion pairs of carbanions. IV. Specific solvation of alkali ions by polyglycol dimethyl ethers. *J. Am. Chem. Soc.* **89**(17), 4547-4549, (1967).

2. Chan, L. L., Wong, K. H., and Smid, J. Complexation of lithium, sodium, and potassium carbanion pairs with polyglycol dimethyl ethers (glymes). Effect of chain length and temperature. *J. Am. Chem. Soc.* **92**(7), 1955-1963, (1970).

4. • Line 107: How was the ‘average voltage’ determined? From Fig 1b it seems that the ‘average Na co-intercalation voltage’ coincides with the onset potential of the sharp intercalation peak, at least for the G1, G2 and G4 electrolytes. For the G_{n-250} and G_{n-500} this, however, does not seem to be the case. A good way to determine the ‘average voltage’ is to determine the energy density E by integrating the voltage-profile over the specific capacity Q. The average voltage V then can be calculated according to $V=E/Q$. Can the authors please comment on the determination of the ‘average voltage’ and whether it is just coincidence that the average voltage appears to be the onset potential of the sharp reduction peak in the G1, G2 and G4 electrolytes?

Reply: We are grateful to the reviewer for the constructive suggestion. Accordingly, we calculated the average voltages based on the method suggested by the reviewer. It shows that the average voltages are about 0.62, 0.65, 0.74, 0.75 and 0.79 V for G₁, G₂, G₄, G_{n-250} and G_{n-500} electrolytes, respectively. These value are close to the 0.60, 0.66, 0.77, 0.78 and 0.81 V for the corresponding electrolytes determined by dQ/dV curves in Fig. 1b. In both cases, there is an increase of the average co-intercalation voltage as the average chain length of the ether increases. It means that the method of evaluating average voltage from the dominant redox peaks in dQ/dV curves is reliable. Note that this method is widely used in literature (*Energy Environ. Sci.* 2015, 8, 2963; *J. Phys. Chem. C* 2018, 12, 26816, and *Angew. Chem. Int. Ed.* 2016, 55, 3129).

As a response to the reviewer’s valuable comment, we add the below sentence in Line 110:

“Note that the above average co-intercalation voltage is determined from dQ/dV curves by the dashed lines located between discharge and charge plateau potentials (Fig. 1b), similar with

previous studies.^{16,17,20} Also note that the average co-intercalation voltages are consistent with the values calculated by the method of dividing the energy density to specific capacity.”

5. • Line 112 and following: It is discussed that the Na-ions are more efficiently shielded by longer solvents, which weakens the interaction with the graphite lattice. For my own curiosity: Is there any experience, whether the rate capability improves with longer ether solvents?

Reply: We would like to thank the reviewer for raising this interesting question. Accordingly, we conducted rate tests of graphite electrodes at 1 M G_1 , G_2 , G_4 , G_{n-250} and G_{n-500} -based electrolytes. It shows that the rate capabilities for graphite electrodes in long ether electrolytes (*i.e.*, G_{n-250} and G_{n-500}) are inferior to those in shorter ether electrolytes (*i.e.*, G_1 and G_2), see Fig. R1a. The poor reaction kinetics can be attributed to the higher viscosity of long ethers and the larger charge transfer resistance (R_{ct}) of cells with long ether electrolytes (Fig. 1Rb). It is noted that viscosities for G_1 , G_2 , G_4 and G_{n-250} solvents were reported to be 0.455, 2.16, 4.05, and 7 cP, respectively (*Electrochimica Acta* 2013, 89, 737).

Figure R1 (a) Rate capacities of graphite electrodes at 1 M G_1 , G_2 , G_4 , G_{n-250} , and G_{n-500} NaPF_6 electrolytes, (b) Nyquist plots of fresh cells with above five electrolytes.

6. • Supplementary Fig 7: Why the reversible capacity of graphite with the 3 M LiPF_6 EC/DEC electrolyte at 25°C is such low? Is this result reproducible? Is there any explanation?

Reply: We are grateful to the reviewer for the careful review. We conducted the experiment several times to ensure the reproducibility of the result. The low capacity of graphite in Supplementary Fig. 7 is probably caused by the high viscosity, low ionic conductivity and poor

electrode wettability of 3 M LiPF₆ EC/DEC electrolyte at room temperature (*J. Electrochem. Soc.* 2005, 152 (5) A882-A891). When we increased the concentration to 4 M, we found that the reversible capacity became lower (*i.e.*, 28 mAh g⁻¹ for 4 M LiPF₆ EC/DEC vs 51 mAh g⁻¹ for 3 M LiPF₆ EC/DEC), supporting above hypothesis. For a fair comparison, we cycled graphite in 2 M LiPF₆ EC/DEC, which shows a high reversible capacity and apparent dQ/dV peaks.

The Supplementary Fig.7 is modified accordingly:

Supplementary Figure 7 (a) Discharge/charge profiles of graphite anodes cycled in 1 M and 3 M (or 2 M) LiPF₆ EC/DEC electrolytes at different temperatures of 25 °C, 35 °C, 45 °C and 60 °C, respectively, (b) the corresponding dQ/dV curves from (a) to show the average Li intercalation potential in graphite anodes under different conditions. Note that we used 2 M electrolyte for the comparison at 25 °C. That is because graphite presented very low reversible capacity and illegible dQ/dV peaks in 3 M electrolyte, possibly due to its high viscosity and poor electrode wettability.

7. • Line 270 and Fig 4a, g: The scale bars of the coulombic efficiencies are not very useful, as one cannot see, how close the coulombic efficiency is to 100%. Can the authors please magnify the scale, maybe from 90% to 105%? It would also be helpful for the reader to mention an exact number of the coulombic efficiency in the text after e.g. 100 cycles instead of ‘close to 100%’.

Reply: We are grateful to the reviewer for the constructive suggestion. The scale bars of Coulombic efficiencies in Figs. 4a and g are magnified accordingly. The following sentence in Line 270 is also modified:

“The Coulombic efficiencies after 10 cycles are about 99.99% (inset Fig. 4g),”

Figure 4 | Electrochemical performance of Na ion full cells. **a**, first ten cycles discharge/charge profiles, and cyclic capacities, Coulombic efficiencies for $\text{Na}_{1.5}\text{VPO}_{4.8}\text{F}_{0.7}$ cathodes for 150 cycles in 2M G_1 -based electrolyte at 100 mA g^{-1} ;**g**, long term cycling performance of Na ion full cells at 1 A g^{-1} for 1000 cycles.

8. • Line 296: I do not fully agree with ‘the absent SEI layer on graphite surfaces’. As discussed above, this question must be addressed by more detailed studies, which are not the scope of this manuscript. Certainly, the SEI film must be very thin, sustain large volume changes and have a minor influence on the co-intercalation kinetics, but I would not call it ‘absent SEI layer’. A

vaguer wording such as ‘... and the beneficial properties of the thin SEI layer...’ might solve this issue.

Reply: We agree with the reviewer’s comment that “the absent SEI layer on graphite surface” is inappropriate.

The following sentence in Line 296 is rephrased accordingly:

“possibly due to the fast reaction kinetics of co-intercalation²² and the beneficial properties of the very thin and stable SEI layer on graphite surface (Supplementary Fig. 15)⁵².”

The following information about SEI on cycled graphite is added in Supplementary Information;

Supplementary Figure 15 TEM and XPS analyses of SEI layer on graphite electrodes after 5 cycles in 2 M NaPF₆ G₁ electrolyte. TEM image indicates that no noticeable SEI layer was formed at the surface of graphite. The XPS results with depth profiling (from surface to 20 nm) also clearly show that the amount of SEI on cycled graphite is rather trivial.

9. • Line 332: Please add the units for the molar masses...

Reply: The units for the molar masses are added in Line 332.

“G₁, G₂, G₃, G₄, G_{n-250} (Mw ≈250 g mol⁻¹) and G_{n-500} (Mw ≈250 g mol⁻¹) solvents at different concentrations in the glovebox.”

10. • General: Sometimes the unit after a value is separated by a space, sometimes not. Please use a consistent style.

Reply: We are grateful to the reviewer for the careful review. All the units after values are now separated by a space. The following sentences are modified accordingly:

Line 154: “slat drastically decrease from 142 (at 0.05 M) to 4.7 (at 1.5 M) and 2.3 (at 3 M).”

Line 156: “concentrated electrolyte above 1.5 M.²⁹”

Line 160: “chemical shift from -8.11 to -6.39 ppm when the concentration increased from 0.05 to 3 M.”

Line 170: “However, in the high concentration region from 1 M to 3M,”

Line 185: “from 0.1 to 2 M, NaClO₄ dissolved in G₂ from 0.5 to 3 M and NaCF₃SO₃ dissolved in G₂ from 0.1 to 2 M.”

Line 356: “treated at 850 °C for 2 h in Ar atmosphere.”

11. • Typo in Supplementary Table 1: Unit of the energy density should be ‘2.5 Wh kg⁻¹’ instead of ‘2.5 Wk kg⁻¹’ in the line of reference 7.

Reply: We need to thank the reviewer again for the careful review. The typo in Supplementary Table 1 is corrected accordingly.

One major issue should also be addressed before acceptance:

12. • Fig 2 and Supplementary Fig 5: The concentration dependent potential shift of the co-intercalation potential might also simply be caused due to the change of the reference potential of the Na-electrode. A way, how one could experimentally rule out, whether the observed effect is an artefact due to a flawed reference electrode is to add a certain amount of ferrocene to the electrolyte and use the ferrocene peak as a reference. I would suggest to perform this experiment for the 0.05 M, the 1 M and the 3 M electrolytes.

Reply: We appreciate this constructive suggestion from the reviewer. Accordingly, we measured the cyclic voltammetry (CV) of co-intercalation reactions of graphite and the ferrocene/ferrocenium (Fc/Fc⁺) redox couple in 0.05 M, 1 M and 3 M NaPF₆ G₂ electrolytes

containing 10 mM ferrocene. In this experiment, we used a three-electrode system: graphite cast on Ni mesh as the working electrode, Pt wire as the counter electrode and Na metal as the reference electrode (*J. Electrochem. Soc.* 2017, 164 (12) A2295-A2297). It is noted that Fc^+/Fc with the best known solvent independent redox system was recommended as a stable internal reference (*Pure & Appl. Chem.*, 1984, 56 (4) 461-466). Assuming the potential of Fc^+/Fc is not affected in the G_2 -based electrolytes, the average co-intercalation voltages were determined to be -2.62, -2.65 and -2.72 V vs. Fc^+/Fc for 0.05 M, 1 M and 3 M, respectively (Supplementary Fig. 19b). The negative shift of co-intercalation potentials versus Fc^+/Fc internal reference is consistent with the decrease of average co-intercalation voltages versus Na^+/Na , as schematically shown in Supplementary Fig. 19c. Therefore, it is safe to state that the average co-intercalation voltage would downshift as increasing the electrolyte concentrations. In fact, in our original Supplementary Information, we tried to explain the shift of average co-intercalation voltage by classic thermodynamics. It was concluded that the activity change of free G_2 molecules determined the co-intercalation voltage change, $\Delta V = \frac{2.303RT}{F} \log \frac{a_{\text{G}_2,3\text{M}}}{a_{\text{G}_2,0.05\text{M}}}$, where R, T, F are gas constant, absolute temperature and Faraday constant, $a_{\text{G}_2,0.05\text{M}}$, $a_{\text{G}_2,3\text{M}}$ are activities of G_2 in 0.05M and 3 M, respectively.

As a response to the Reviewer's suggestion, the following figure and references are added in Supplementary Information:

Supplementary Figure 19 (a) Cyclic voltammetry curves of graphite electrode cycled in 0.05 M, 1 M and 3 M NaPF₆ G₂ electrolytes containing 10 mM ferrocene. The experiments were conducted in three electrode systems using graphite cast on Ni mesh as working electrodes, Pt wire as counter electrodes and Na metal as reference electrodes³. The scan rate is 1 mV s⁻¹. (b) the average co-intercalation voltages vs. Fc⁺/Fc as a function of electrolyte concentrations, (c) schematically showing the left-shift of co-intercalation potential versus Na⁺/Na and Fc⁺/Fc references. Considering Fc⁺/Fc the best known internal reference,⁴ the average co-intercalation voltages vs. Fc⁺/Fc are regarded negative shift as increasing electrolyte concentrations, confirming our finding about the concentration dependent shift of the co-intercalation voltage. Note that the Fc⁺/Fc redox couple in 0.05 M electrolyte is enlarged to index due to the large polarization in the extremely dilute electrolyte.

3. Mozzhukhina, N. and Calvo E. J. The correct assessment of standard potentials of reference electrodes in non-aqueous solution, *J. Electrochem. Soc.* **164** (12) A2295-A2297 (2017).

4. Gritzner, G. and Kuta, J. Recommendations on reporting electrode potentials in nonaqueous solvents. *Pure Appl. Chem.* **56**, 464–466 (1984).

REVIEWERS' COMMENTS:

Reviewer #1 (Remarks to the Author):

The authors have addressed the concerns and comments to a very large extent. I would like to suggest that this version be accepted.

Reviewer #2 (Remarks to the Author):

The manuscript revision is appreciated and recommended for publication.

Lukas Seidl

Authors' response to Reviewers' Comments:

The authors appreciate very much the invaluable comments offered by the reviewers and the editor.

Reviewer #1 (Remarks to the Author):

The authors have addressed the concerns and comments to a very large extent. I would like to suggest that this version be accepted.

Reply: We are grateful to the reviewer's efforts in reviewing our work.

Reviewer #2 (Remarks to the Author):

The manuscript revision is appreciated and recommended for publication.

Lukas Seidl

Reply: We really appreciate the valuable comments provided by the reviewer in improving the quality of our manuscript.